# Does DNA Methylation Matter in FSHD?

**DOI:** 10.3390/genes11030258

**Published:** 2020-02-28

**Authors:** Valentina Salsi, Frédérique Magdinier, Rossella Tupler

**Affiliations:** 1Department of Life Sciences, University of Modena and Reggio Emilia, 4, 41121 Modena, Italy; valentina.salsi@unimore.it; 2Aix Marseille Univ, INSERM, MMG, U 1251, 13005 Marseille, France; Frederique.MAGDINIER@univ-amu.fr; 3Department of Biomedical, Metabolic and Neural Sciences, University of Modena and Reggio Emilia, 4, 41121 Modena, Italy; 4Center for Neuroscience and Neurotechnology, University of Modena and Reggio Emilia, 4, 41121 Modena, Italy; 5Department of Molecular Cell and Cancer Biology, University of Massachusetts Medical School, Worcester, MA 01003, USA; 6Li Weibo Institute for Rare Diseases Research at the University of Massachusetts Medical School, Worcester, MA 01003, USA

**Keywords:** FSHD1, FSHD2, D4Z4 macrosatellite, DNA methylation, epigenetics

## Abstract

Facioscapulohumeral muscular dystrophy (FSHD) has been associated with the genetic and epigenetic molecular features of the CpG-rich D4Z4 repeat tandem array at 4q35. Reduced DNA methylation of D4Z4 repeats is considered part of the FSHD mechanism and has been proposed as a reliable marker in the FSHD diagnostic procedure. We considered the assessment of D4Z4 DNA methylation status conducted on distinct cohorts using different methodologies. On the basis of the reported results we conclude that the percentage of DNA methylation detected at D4Z4 does not correlate with the disease status. Overall, data suggest that in the case of FSHD1, D4Z4 hypomethylation is a consequence of the chromatin structure present in the contracted allele, rather than a proxy of its function. Besides, CpG methylation at D4Z4 DNA is reduced in patients presenting diseases unrelated to muscle progressive wasting, like Bosma Arhinia and Microphthalmia syndrome, a developmental disorder, as well as ICF syndrome. Consistent with these observations, the analysis of epigenetic reprogramming at the D4Z4 locus in human embryonic and induced pluripotent stem cells indicate that other mechanisms, independent from the repeat number, are involved in the control of the epigenetic structure at D4Z4.

## 1. Introduction

Facioscapulohumeral muscular dystrophy (FSHD, OMIM#158900) is the third most common myopathy with a reported prevalence ranging between 1 in 8333 [1] and 1 in 20,000 [2]. FSHD is characterized by insidious onset and progressive wasting of a highly selective set of muscle groups, the facial, limb girdle, and foot extensor muscles, and by a great variability of clinical expression among patients and within families [3,4]. The disease appears significantly earlier in males [5,6,7] and this determines male patients to be in higher number and more severely affected than females [5,6,8,9,10]. In some families, individuals affected by FSHD can be found only in one generation [5,7,11,12]. These differences are striking in discordant monozygotic twins [13,14,15].

Genetically, FSHD has been considered a Mendelian disease with autosomal dominant inheritance twins [16,17,18]. On this basis, the FSHD genetic locus was mapped on chromosome 4q35 by genetic linkage analysis [16,18,19] and associated with rearrangements of an array of tandemly repeated 3.3 kb segments (D4Z4) [20]. FSHD is the only human disease causally linked to copy number variation of macrosatellite DNA elements [21]. The number of D4Z4 repeats varies from 11 to 100 in the general population, whereas 10 repeats or fewer are usually found in sporadic and familial FSHD patients. As a general rule, alleles composed by 11–100 copies of D4Z4 repeats constitute the normal size range of D4Z4 alleles, whereas alleles with 8 or fewer D4Z4 repeats are considered diagnostic for the disease [22,23].

The routine DNA molecular testing, based on the identification of D4Z4 arrays with less than 10 units at 4q35, has been considered highly sensitive and specific [23,24]. However, several genotype–phenotype studies have shown that D4Z4 reduction (4–8 reduced units, RU) is also present in 3% of the general population [25,26,27,28] and in some cases is associated with distinct myopathic phenotypes not reminiscent of FSHD [29,30]. Moreover, 10% of FSHD patients carry D4Z4 alleles of borderline size (9–10 RU), which are also found in healthy people or in subjects with a different myopathy [31] (Ricci et al., submitted). Finally, 5–10% of FSHD patients carry D4Z4 arrays of size within the range of the general population (11 RU or more) on both chromosomes 4q [32,33]. These subjects represent a second form of disease, FSHD2. Therefore, the number of D4Z4 RU at 4q35 does not per se characterize FSHD (Figure 1).

As a matter of fact, genotype–phenotype studies have shown a large spectrum of clinical phenotypes in myopathic subjects [29,30,31], carrying a D4Z4 reduced allele as well as reduced penetrance among relatives carrying the same D4Z4 reduced allele [7,9,10,12,25,34,35]. All this has relevant effects on clinical practice, complicating diagnosis, prognosis, and genetic counseling.

## 2. Molecular Features and the Epigenetic Model for FSHD

The D4Z4 repeat is part of a family of 3.3 kb sequences frequently found in heterochromatic regions, such as the short arms of the acrocentric chromosomes [36], and a nearly identical and equally polymorphic D4Z4 array reside at the sub-telomere of chromosome 10q [37,38].

D4Z4 is a CpG-rich (73%) macrosatellite DNA element encompassing more than 16 nucleosomes and containing multiple repeat sequences normally associated with heterochromatin [36]. The D4Z4 repeat unit contains the open reading frame of the retrogene *DUX4*.

Studies showed that D4Z4 plays a role in control of gene silencing at 4q35 through the recruitment of a multi-protein repressor complex, the D4Z4 Recognition Complex (DRC), composed of YY1, HMGB2, and Nucleolin, binding a 27bp DNA element within the D4Z4 sequence (D4Z4 Binding Element, DBE) [39]. The factor YY1 is a component of multiple chromatin regulatory complexes, including the Polycomb Repressive Complex 2 (PRC2), which includes the H3K27 methyltransferase EZH2 [40]. YY1 also interacts with histone deacetylase 1 (HDAC1), HDAC2, and HDAC3 [41], as well as with PARP1 [42]. A long non coding RNA, DBE-T, encoded within the 4q35 locus, is selectively transcribed in FSHD samples and participates in the transcriptional and epigenetic regulation of the 4q35 genes [43].

Several clinical features, such as penetrance variability, gender bias in severity [6], asymmetric muscle wasting, and discordance in monozygotic twins [13,14,15], suggest that FSHD development involves epigenetic factors which might influence gene expression through local modification of chromatin structure.

Indeed, the great clinical heterogeneity and molecular uncertainty has pointed to the need for additional markers to support FSHD diagnosis, genetic counseling, and patient stratification for clinical trials.

In the rarer cases of FSHD2 (OMIM#158901), more than 80% of subjects carry heterozygous mutations in the *SMCHD1* (structural maintenance of chromosomes flexible hinge domain containing 1) gene, which encodes a chromatin remodeling protein required for normal DNA methylation levels and transcriptional repression at certain loci, including the inactive X chromosome, imprinted genomic regions, and the D4Z4 arrays [44,45,46]. A few FSHD2 cases carry heterozygous mutations in the *DNMT3B* (DNA methyltransferase 3 beta) gene [47] which is responsible for the establishment of the proper de novo cytosine methylation profile during development. FSHD1 and, at a greater extent, FSHD2 patients present D4Z4 DNA hypomethylation [33,48,49].

All these findings supported the hypothesis that significant alteration of the 4q35 epigenetic landscape is central to FSHD, and that this occurs in FSHD1 by removal of a significant number of D4Z4 heterochromatic elements, and in FSHD2, via SMCHD1 or DNMT3B haploinsufficiency in presence of a specific telomeric polymorphism, 4qA, which allows the expression of the most distal copy of the DUX4 gene. More specifically, the PAS present on 4qA alleles allows for DUX4 mRNA polyadenylation, stabilization, and translation [50,51,52]. In turn, all these alterations have brought forward a unifying model for FSHD pathogenesis, involving the loss of epigenetic silencing and the consequent aberrant expression of the *DUX4* retrogene [52] (Figure 1). At present, these molecular characteristics are used to explain FSHD pathogenesis and to develop therapeutic approaches.

## 3. The Debated Role of DNA Methylation in FSHD: Clinical and Families Studies

DNA methylation, a covalent post-synthetic modification of cytosines engaged in CpG dinucleotides, is a heritable epigenetic mark. In healthy individuals, 70% to 90% of the CpGs are methylated in somatic tissues, representing between 0.75% and 1% of the total number of bases in the diploid human genome [53]. In human cells, dispersed CpGs are methylated whereas CpG clusters (CpG islands) are mostly unmethylated [54]. Repetitive DNA sequences, enriched in CpGs, are usually packed as condensed and repressed chromatin by dense methylation, a protective mechanism that inhibits the invasion of the genome by the reactivation of transposable elements [55,56]. DNA methylation plays important roles in a number of physiological processes, such as development [57,58] and ageing [59,60,61]. Aberrant DNA methylation plays a causal role in a variety of diseases, including cancer [62,63]. In general, cancer cells exhibit DNA hypermethylation of promoter regions of tumor suppressor genes and global hypomethylation of repetitive DNA sequences accompanied with an increased genomic instability or loss of heterozygosity. Methylation changes also contribute to a number of diseases [64,65]. In particular, methylation have been investigated in diseases associated with short tandem repeats DNA [66] and CpG methylation has been considered a possible disease marker or modifier [67]. In fact, DNA hypermethylation is a critical feature of expanded short repeat arrays, as reported for Fragile X syndrome [66] or Friedreich’s ataxia [68].

In FSHD, reduction of epigenetic silencing and aberrant gene expression seems to constitute the underlying mechanism leading to disease. It has thus been thought that reduced methylation at the D4Z4 locus might represent a proxy indicator of the reduced transcriptional silencing associated with FSHD. As a consequence, low CpG methylation of the D4Z4 sequence has been proposed as a reliable marker in the FSHD diagnostic procedure [33,48,49].

To investigate this hypothesis on a large-scale, different works have been conducted on distinct clinical cohorts. These studies assessed the methylation status at D4Z4, trying to correlate the percentage of CpG methylation at the D4Z4 array with one or more FSHD features [69,70,71,72,73].

In early works, DNA methylation has been investigated by Southern blotting after digestion with methylation-sensitive restriction enzymes (MSRE assay) [33,48] and hybridization with the p13E11 probe, which detects a unique region upstream of the first repeat of the D4Z4 array (Figure 2). The percentage of CpG methylation of the D4Z4 was estimated based on the density of the hybridization signals of restriction fragments obtained prior or after restriction digestion with MSRE. The most proximal *Fse*I site within the D4Z4 array was considered as the most sensitive for this assay with an estimated DNA methylation of 70–80% in healthy individuals and a significant decrease in FSHD1 patients. The MSRE assay possesses a restricted range of investigation since it evaluates only the few CpGs detected by the various of MSREs without distinction between the D4Z4 repeat array on the 4q and 10q chromosome [69], thus offering a narrow view of the D4Z4 methylation status.

To overcome these limitations, the bisulfite treatment of DNA followed by DNA sequencing (BSS) has become the technique of choice to assess the D4Z4 methylation, virtually allowing the detection of the methylation status of any CpGs within the array at once.

With this technical improvement, many works succeeded in reporting a more exhaustive analysis of many CpGs within all D4Z4 units, even if without distinction between the 4q and 10q alleles [70,71]. Recently, Jones et al. [69] developed a 4q- and 10q-specific protocol for BSS analyses, which is concomitantly able to capture the epigenetic status of the 3′ end of the most distal D4Z4 repeat and the abutting A-type locus, which has been correlated with the pathogenic expression of *DUX4* on short D4Z4 arrays. A further advance in the methodology used in methylation studies was obtained by Roche et al. [74] consisting in the assessment of DNA methylation at D4Z4 by using a custom deep sequencing method after sodium bisulfite conversion of genomic DNA.

Nevertheless, the different attempts to use BSS analysis to assess D4Z4 epigenetic status suffer from several limitations, both from a conceptual and a technical point of view.

As shown in Table 1, we compared the results of the main reports in the field. The comprehensive analysis of these studies highlights some problematic points: (1) the lack of a clear description of the clinical status of individuals within each cohort [69,70,71,72,74]; (2) the ambiguous definition of asymptomatic or healthy carriers of a D4Z4 allele of reduced size, since they are included either in the group of people with FSHD even if their clinical score is equal to zero, or in the control group despite their liability to develop the disease [69,72]; (3) the constant attempt to correlate the methylation status with the disease severity [70,72]; and (4) the effort to uniquely correlate hypomethylation at D4Z4 with *DUX4* expression [69,70,72].

From a technical point of view, the scenario is even more complicated by the fact that each work used different primers sets in the BSS analyses to evaluate CpGs within different regions of the D4Z4 array (Figure 2) and applied different statistical tools. First, for a proper analysis of DNA methylation by bisulfite sequencing, (1) primers should not contain CpG sites within their sequence to ensure unbiased amplification of both methylated and unmethylated DNA; and (2) primers should be designed in a region containing enough number of non-CpG cytosines to avoid amplification of incompletely modified genomic DNA [75]. Despite these recommendations in some of the works considered here, the primer designed for BSS analysis included CpGs [69] or just one or few non-CpG cytosines [73], which might affect the amplification of the converted DNA. Secondly, the selection of different regions within the array led to controversial results about which is/are the region/s more representative for the methylation status at D4Z4.

As anticipated, we also observed that statistical analyses performed in each individual study could not be compared with each other. For instance, in the work by Calandra et al. [72], the assessment of the correlation between methylation levels and disease was performed focusing on a single CpG out of ten within a region where D4Z4 hypomethylation is more evident.

In the study by Lemmers et al. [70] the global methylation levels were estimated by means of a novel statistical model considering the D4Z4 array as a linear string of mathematic units: the methylation level was calculated as a linear function of D4Z4 repeats number. In biological terms, this arbitrary approach has an intrinsic bias, since the repeat length of each allele affects the 3D chromatin structure of the individual D4Z4 locus, and most likely its methylation status. Longer alleles generate a condensed chromatin structure and beyond a certain number of repetitive elements, the D4Z4 region displays a high percentage of CpG methylation irrespective of the discrete D4Z4 repeat number of each individual long allele.

Moreover, when considering alleles with fewer D4Z4 repeats, the effect on DNA methylation differs in each D4Z4 allele and depends on the individual size. In fact, in families in which it was possible to follow the segregation of D4Z4 alleles analyzed by bisulfite sequencing, it was possible to discriminate between the two D4Z4 alleles carrying different repeat numbers. In these cases, the degree of methylation roughly correlates with number of alleles with 10 or fewer repeats, but it was not associated with the clinical condition [73].

Regardless of the methodologies used, data indicates that D4Z4 methylation is highly variable also in healthy individuals and that hypomethylation concerns only a limited number of CpGs within the D4Z4 sequence. Hypomethylated CpGs are mostly clustered in the proximal part of the repeat but not in the *DUX4* promoter region per se or in the distal part of the coding sequence and 3′ UTR [69].

Overall, the observed hypomethylation of D4Z4 in FSHD patients shows a correlation with the reduction of D4Z4 repeat units, even though the accurate review of the published data reveals a lack of association between D4Z4 methylation level and FSHD1 clinical status.

In conclusion, a detailed phenotypic characterization should be recommended, together with a common study design and a consensus approach in order to reach a more accurate and unbiased evaluation of the global methylation pattern and to draw any conclusion on its clinical significance.

## 4. DNA Methylation in FSHD2

The SMCHD1 chromatin-associated factor has been implicated in FSHD on the basis of Whole Exome Sequencing of 16 FSHD2 families presenting residual D4Z4 methylation below a threshold of 25% at the most proximal *Fse*I site [76]. It is now recognized that approximately 80% of FSHD2 patients are carriers of one *SMCHD1* variant [77]. In a subsequent study, the threshold that defines hypomethylation was estimated to be 30% of residual methylation at this proximal *Fse*I site [70]. However, there seems to be no strict correlation between *SMCHD1* variants, D4Z4 hypomethylation, and the appearance of clinical signs. This is exemplified by the non-affected individuals who display a low DNA methylation level <30%; 26% for Rf854-case 2398; 30% for Rf854-case 2434; and 29% for Rf300-Cases 3 and 4, presented in the original work of Lemmers et al. [76]. More strikingly, considering the clinical phenotype, the marked D4Z4 hypomethylation detected in the majority of FSHD2 patients carrying a mutation in *SMCHD1* is not accompanied either with a more severe phenotype or earlier disease onset. Moreover, D4Z4 hypomethylation, in the presence of the haplotype 4qA/PAS distal to the D4Z4 array, which is considered permissive of *DUX4* expression, is not associated with a muscle phenotype in patients with Bosma Arhinia and Microphthalmia syndrome (BAMS), patients affected with ICF (Immunodeficiency, Centromeric Instability, and Facial anomalies) homozygotes for *DNMT3B* mutation and their heterozygote parents [78], or patients carrying a deletion of the 18p chromosome containing the *SMCHD1* gene [77]. Besides, in the few families with FSHD index cases carrying a mutation in *DNMT3B* [47] or an 18p deletion [79,80], the presence of a DNA mutation and D4Z4 hypomethylation does not segregate with clinical signs of the disease. Furthermore, as highlighted by molecular combing [81], a significant proportion of individuals affected by FSHD display an atypical genotype, with complex distal rearrangements [82], presence of proximal deletions, additional 10q alleles, in the absence of D4Z4 array shortening, *SMCHD1* variants, or D4Z4 hypomethylation [83].

So far, further investigations are required to answer a few important questions, such as the role of epigenetic alterations in FSHD, whether hypomethylation leads to *DUX4* activation and, above all, how this translates into a specific muscle phenotype.

## 5. Changes in DNA Methylation at D4Z4 Upon Reprogramming

Pluripotent stem cells represent an important tool to model human genetic disorders, as they can be used to analyze the effect of specific genomic alterations in early development or during differentiation [83,84,85,86]. Cell reprogramming always involves profound epigenetic changes and the acquisition of an epigenetic pattern similar to that of pluripotent embryonic stem cells. The pattern of DNA methylation is not fully erased after reprogramming. Hence, reprogrammed cells keep memory of the pattern of the parental somatic cells they were derived from, but also acquire DNA methylation profiles specific of induced pluripotent stem cells (iPSCs). iPSCs obtained by reprogramming of patients’ cells have been recently used to clarify the molecular basis of several “epigenetic” diseases. For instance, in human hiPSCs from Fragile X (FXS) patients, the *FMR1* promoter region aside the 5′UTR with expanded CGG triplets (n > 200) remains in most cases hypermethylated, suggesting that once established in FXS fibroblasts, these epigenetic marks are stably maintained after reprogramming, whereas in other cases, the methylation pattern of the disease-associated locus does not reflect the profile of the donor cells [87,88,89,90].

Concerning FSHD1, the investigation of DNA methylation at D4Z4 in iPSCs from affected subjects and healthy donors revealed that these cells do not retain the methylation pattern inherited from the donor cells, but acquire a new methylation profile. This argues for the presence of two mechanisms acting at D4Z4 upon reprogramming: an active “erasure” of the cell-of-origin epigenetic profile and a “rewriting” of a de novo methylation pattern at the array. Moreover, reports demonstrate that the D4Z4 methylation was identical from clone to clone from both FSHD1 patients and controls, indicating that remethylation of D4Z4 upon epigenetic reprogramming does not depend on the residual number of D4Z4 repeats [79].

More in details, the comparison of the methylation profile between FSHD iPSCs and human Embryonic Stem Cells (hESCs) with a short or long D4Z4 array, revealed a trend toward a high methylation level, without any significant difference between FSHD1 and control cells and between iPSCs and ESCs.

The higher methylation level found at D4Z4 in pluripotent cells further suggests that D4Z4 methylation status does not correlate with the number of repeats but is a feature of stemness, which highlights once more the complex but yet unknown regulatory mechanisms of this locus [79].

Interestingly, recent findings revealed a key role for *DUX4* at very early stages of human development and the activation of embryonic genes at the zygotic genome activation stage (ZGA) [91,92,93,94], a stage characterized by profound methylation changes. A high level of methylation was observed in hiPSCs and hESCs. These cells represent a later developmental stage compared to ZGA since they are derived from the blastocyst inner cell mass (ICM), a developmental stage characterized by remethylation of the zygotic genome after post-fertilization waves of demethylation up to the morula stage. These findings rule out the connection between *DUX4* expression and D4Z4 hypomethylation.

Additional features, such as the size of the abutting telomere, might be also implicated in the regulation of D4Z4 methylation. Interestingly, the highest level of methylation is observed in cells where telomerase is reactivated, such as hiPSCs and ES cells [95,96,97]. These observations suggest the possible link between D4Z4 methylation and telomeres and also draw attention to the complexity of the epigenetic regulation of this macrosatellite element during development.

## 6. Trans-Acting Factors

The possibility to use the percentage values of methylated CpGs at D4Z4 as a diagnostic marker for FSHD is further weakened by the fact that DNA methylation could not be considered per se. It is now well established that DNA methylation is only one among numerous indicators of chromatin structure at D4Z4 and several research groups reported that alterations of chromatin structure is reflected by post-translational histone modifications [48,98,99,100], by the binding of various non-histone proteins and RNAs on the repeat array [39,43,101], and by higher order chromatin structures formation [102,103,104].

Epigenetic alteration of chromatin relies on DNA methylation and on histone tails modifications in nucleosomes, which together determine the so called “epigenetic signature” of a specific region. It has been reported that the D4Z4 array contains histone H3 Lysine 9 trimethylation (H3K9me3), a repressive mark associated with heterochromatin formation, together with H3K27me3, a repressive chromatin mark associated with Polycomb silencing [33,43,98]. A specific loss of H3K9me3 followed by the loss of HP1γ and cohesin binding at D4Z4, suggestive of a more relaxed chromatin structure [98], was reported both for FSHD1, in the presence of a reduced number of repeats, and also in FSHD2 subjects. The loss of repressive histone marks is not a direct consequence of DNA hypomethylation since H3K9me3 enrichment at D4Z4 was unaltered in ICF syndrome, which displays severe DNA hypomethylation at the 4q35 locus [49,98]. Besides, H3K27 presence at D4Z4 did not show any changes between the control and FSHD muscle cells [43,102].

Other observations argue for a role for the Polycomb proteins in the epigenetic regulation at D4Z4: The methyl-transferase EZH2 (Enhancer of zeste homolog 2), a component of Polycomb repressive complex 2 (PRC2), was shown to be reduced in FSHD muscle cells, while ASH1L (ASH1 Like Histone Lysine Methyltransferase), a member of the Trithorax complex associated with transcriptionally active chromatin, was found to be specifically recruited to D4Z4 in FSHD cells [43]. Notably, ASH1L recruitment has been shown to be dependent on the expression of the non-coding RNA DBE-T, transcribed by to the most proximal D4Z4 unit. In particular, DBE-T-mediated ASH1L recruitment to D4Z4 in FSHD cells is reported to mediate histone H3 lysine 36 dimethylation (H3K36me2), a major histone mark associated with transcriptional activation, leading to chromatin remodeling and 4q35 gene transcription [43].

Among transcription factors, two proteins have mainly been associated with D4Z4, YY1 Yin Yang 1) and CTCF (CCCTC-binding factor). Reports demonstrated the association of a repressor protein complex, named DRC (D4Z4 Recognition Complex) composed of YY1, a known transcriptional repressor, HMGB2 (high-mobility group protein 2), an architectural protein, and nucleolin, with the 27 bp DBE (D4Z4 binding element) sequence contained within each D4Z4 [39]. YY1 was reported to be the major factor bound to the DBE, although other DRC components contributed to DNA binding affinity and specificity. It is noteworthy that YY1 binds DNA only when it is not methylated [105]. The region encompassing the DBE element is hypermethylated in FSHD and control subjects, except from the CpGs neighboring the YY1 consensus site, which show variable levels of methylation (Tupler, personal observation and [106]). Conversely, CTCF binding is disrupted by CpG methylation [107]. Two CTCF binding sites are located within the 5′ region of D4Z4 repeat, which is reported to display hypomethylation in the presence of a DRA [74,101,106,108]. CTCF binding to 4q35 displays an inverse correlation to D4Z4 copy number and function as a Lamin A/C-dependent chromatin insulator, protecting D4Z4 from epigenetic silencing by surrounding heterochromatic regions, therefore keeping the D4Z4 chromatin open primarily in a DRA context [102]. Since CTCF has also been shown to mediate chromatin loop formation and to generate TADs (topologically associated domain) [108,109,110,111] the increased CTCF binding to D4Z4 in FSHD may result in altered nuclear and chromatin organization. In particular, using an integrated genome wide approach (4C-seq) it has been reported that 4q-D4Z4 interactome is altered in FSHD1, leading to a chromatin switch toward an active state (mediated by enhancer–promoter interactions), which in turn results in the transcriptional activation of genes involved in muscular atrophy [103].

All these observations further complicate the comprehension of D4Z4 activity regulation and confirm that DNA methylation and its consequence on DUX4 activation are only a part of the fine tuning controlling the transition between healthy and disease status.

## 7. The Significance of D4Z4 Hypomethylation in FSHD and Its Implication for Clinical Counseling

The main current theory about the FSHD pathogenic model involves the aberrant expression of *DUX4* retrogene caused by epigenetic changes in the D4Z4 region [93]. This should be due to reduced methylation, either for D4Z4 repeats reduction in FSHD1 or for the effect of the *SMCHD1* gene in FSHD2 [52,76]. However, the literature reports clearly reveals that hypomethylation correlates with the reduction of the arrayed repeats, but not with the disease status [69,70,71,72]. It is thus conceivable that in FSHD1, hypomethylation is a consequence of the chromatin structure present in the contracted allele, instead of an indicator of its function.

This also holds true for FSHD2, especially for patients who carry mutations in *SMCHD1*, deletions of the 18p locus or mutations in *DNMT3B,* and especially considering the molecular overlaps (D4Z4 hypomethylation) with other unrelated diseases, such as BAMS, ICF, or 18p deletion syndromes (Figure 3).

All these observations indicate that accurate diagnosis should not completely rely on molecular findings. Large studies based on clinical evaluation demonstrate that (1) 2% of healthy individuals from the general population bear one FSHD1-sized D4Z4 allele with the possibility of expressing the DUX4 transcript [27]; (2) the clinical severity of FSHD does not clearly associate with the size of the deletion of the D4Z4 allele [7,12,112]; (3) the penetrance of FSHD is not complete, with 20–50% of relatives carrying FSHD1-sized D4Z4 alleles being healthy [7,9,10,25,34,35]; (4) the disease penetrance decreases among second-fifth degree relatives in FSHD families [7]; (5) there are families showing affected subjects in one generation, families with a single affected subject, and also subjects with atypical phenotypes in which additional mutations in genes causing other neuromuscular diseases have been found [7,30,34]; in addition, independent studies showed that (6) *SMCHD1* mutations cause the Bosma developmental syndrome (OMIM #603457) and D4Z4 hypomethylation has been found in these patients with no documented signs of FSHD [113,114]; (7) *SMCHD1* mutations have been found in subjects with Limb Girdle Muscular Dystrophies (LGMDs) (Tupler, unpublished NGS data); (8) homozygous or compound heterozygous mutations in *DNMT3B* cause immunodeficiency-centromeric instability-facial anomalies syndrome-1 (ICF1) (OMIM #242860), a syndrome characterized by hypo- or agammaglobulinemia and frequent infections that determine poor life expectancy; and (9) some ICF1 cases showed D4Z4 hypomethylation with no signs of FSHD [78] (Figure 3).

Thus, *SMCHD1* and *DNMT3B* mutations have been associated with congenital diseases, whereas FSHD is an adult progressive disease with no congenital presentation, and very importantly, no FSHD patients carrying *SMCHD1* mutations have signs of Bosma syndrome even when they carry overlapping missense mutations or mutations in the same regions of the coding sequence. These data are consistent with the frequent observation that genes play roles in multiple pathways, and that different mutations within the same gene can contribute to distinct phenotypes with different mechanisms of action (Figure 3).

## 8. Conclusions

Unraveling the mechanism leading to FSHD reveals to be a prickly issue. One of the major challenges we face in clinical practice is that there is no unique molecular signature discriminating FSHD patients. We are still trying to promote a linear common diagnostic approach to FSHD, an ambitious objective that should involve all clinical practitioners in the field, as displayed by the use of the FSHD Comprehensive Clinical Evaluation Form [29,115]. At present, current available molecular markers are not sufficient for a proper diagnosis or prognosis, nor for genetic counseling, but patients are still looking for support and assistance on a daily basis. Because of the large phenotypic variations observed among affected subjects, the clinical assessment should be performed carefully considering the family context, with the attempt to find a pattern of inheritance and shared features

Our hypothesis is that FSHD clinical variability may result from the combinatorial effects of variant alleles in genes exerting a detrimental effect on muscle function, together with epigenetic changes influencing their expression. The genetic background of each individual, including harmful gene variants or variants in regulatory elements, might have a crucial role in disease penetrance, high clinical variability and unpredictable progression. In view of the results of genotype–phenotype studies on FSHD cohorts and of the recent findings on the effects of *SMCHD1* mutations, it is advisable to apply a stringent analysis of clinical phenotypes to decipher molecular data. An oversimplification may cause misdiagnosis and biased interpretation of molecular findings with severe effects not only for research, following incorrect hints, but also for clinical practice. The systematic collection of precise phenotypic data of patients and families should become an ineludible tool to support research in human molecular genetics and to translate molecular findings into clinical practice.

Besides, the D4Z4 array and the regulation of its chromatin dynamics has become more and more intriguing over the recent years because of the identification of numerous factors able to regulate its chromatin state and topology within the genome. This element has often been considered separately and not necessarily in the context of the 4q35 locus itself, which also contains several microsatellites of unknown function upstream of D4Z4, and repetitive beta satellite elements distal to the D4Z4, not mentioning the nearby telomere. Variations in macrosatellite number and associated features suggest that not only D4Z4, but also other elements of the large family of repetitive sequences, might play important regulatory functions either locally or at long distances on gene expression. It is speculated that copy number variations of repetitive elements might be associated with variations in human phenotype and they could explain different susceptibility to the disease and missing heritability. In FSHD, the molecular features of the 4q35 sub-telomeric region could be an example of this molecular and clinical variegation.

In conclusion, the study of CpG methylation, performed by several groups and in distinct ways, failed to reveal a clear and unifying theory about the “epigenetic” basis of FSHD. Indeed, the D4Z4 methylation status does not mirror the clinical expression of the disease. The measurement of this epigenetic mark must be interpreted with caution and may support the characterization of individual FSHD families in clinical practice rather than be assumed as a marker of D4Z4 function in FSHD.

## Figures and Tables

**Figure 1 genes-11-00258-f001:**
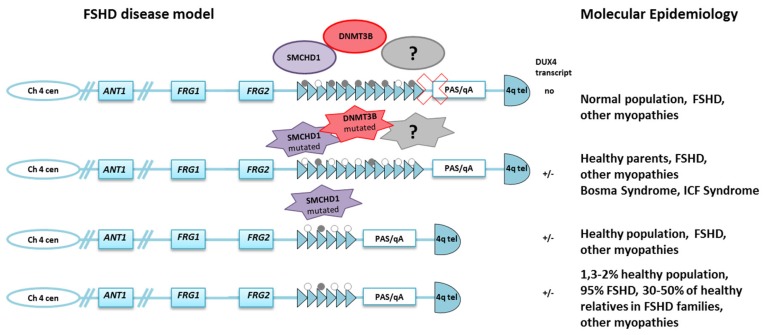
Molecular complexity in facioscapulohumeral muscular dystrophy (FSHD). D4Z4 contractions are associated with a permissive 4qA genotype that involves the aberrant expression of the *DUX4* retrogene and is responsible for FSHD1, but also occurs in 1.3–2% of the normal population. 4qA is also found in cases presenting complex phenotypes and as well as in 30–50% of healthy relatives. Mutations in *SMCHD1* or *DNMT3B* genes have been associated with FSHD2 and are responsible for ICF and BAMS syndromes. Parents of ICF patients are heterozygous for *DNMT3B* pathogenic variants but do not show any sign of muscular dystrophy. Other myopathic patients carry *SMCHD1* mutations. Grey dot = methyl CpG; white dot = unmethylated CpG.

**Figure 2 genes-11-00258-f002:**
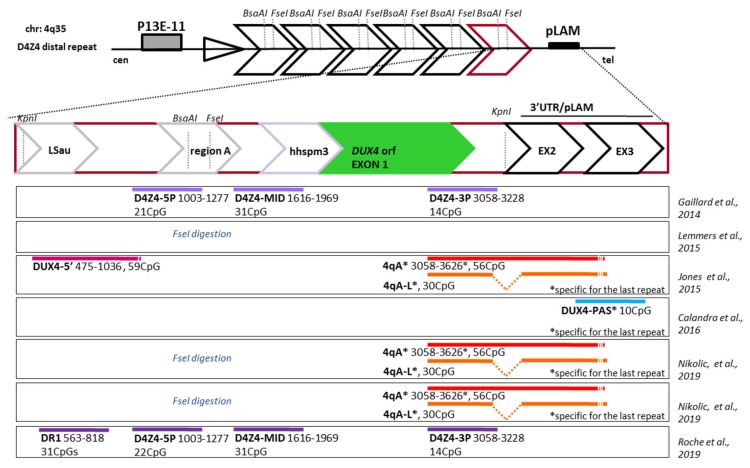
Overview of the D4Z4 repeats and of the region selected in methylation analyses. The D4Z4 array at 4q35 with an enlarged schematic representation of the D4Z4 distal most repeat (from position 1 to 3303 given relative to the two flanking *Kpn*I sites). The different regions within D4Z4 are indicated: LSau repeat (position 1–340), Region A (position 869–1071), hhspm3 (position 1313–1780), *DUX4* ORF (position 1792–3063), plus the 3′ pLAM region. The figure highlights the proximal *Bsa*AI and *Fse*I methylation-sensitive restriction sites analyzed by Southern blotting, and the location of bisulfite (BS) PCR products used in the selected literature reports, each represented with a differently colored line. The position of each region within the array, starting form the first *Kpn*I site, is indicated together with the assayed number of CpGs. The white lines within these regions indicate the presence of CpGs in PCR primers.

**Figure 3 genes-11-00258-f003:**
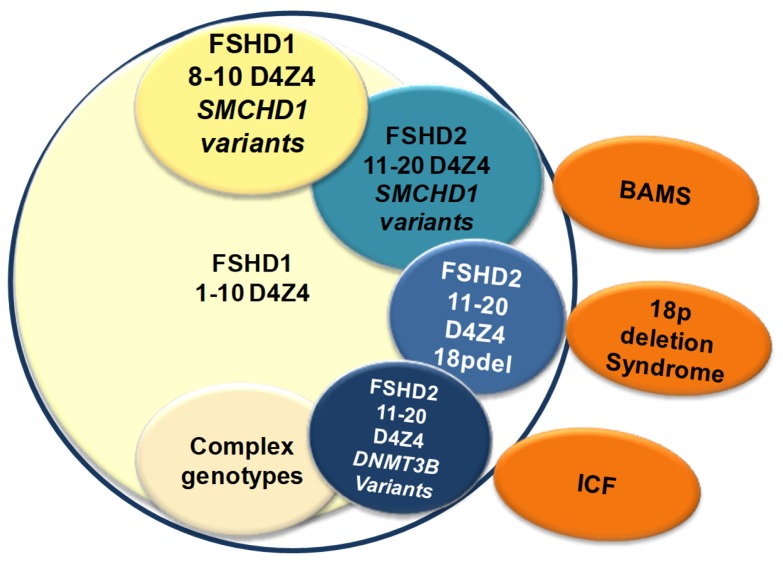
The lack of a unique molecular signature in FSHD complicates genotype–phenotype correlation in clinical practice. The figure illustrates the complex molecular scenario observed in individuals with clinical features of FSHD. Hypomethylation is observed in a large subset of them, including in patients who carry a mutation in *SMCHD1* or *DNMT3B*. This scheme also highlights that hypomethylation of D4Z4 is observed in other rare diseases linked to mutations in *SMCHD1* (BAMS), *DNMT3B* (ICF) or carrying a deletion of the 18p locus that contains the *SMCHD1* gene.

**Table 1 genes-11-00258-t001:** Systematic analysis and comparison of the major literature reports on D4Z4 methylation in FSHD.

REFs	Method	Assayed Region (s)	4q- Specific	Nr.r of Subjects	Clinical Status (CCEF/ACSS)	Conclusions				
						Hypomethylation in FSHD1 /FSHD2	Position within the Locus in which Hypomethylation was Found to be Discriminant	Correlation with D4Z4 Size	Correlation with Disease Severity	Matters
[69]	BSS	4qA; 56 CpGs or 4qAL; 30 CpGs DUX4 5’; 59CpGs	Yes No	Famil:ies 12 Controls 10 Asymptomatic 22 FSHD1	Not reoprted	Yes; Asymptomatyc carriers with intermediate methylation levels.	4qA: pLAM only No DUX4 5’	No	Not tested	.Different results using 5’ primers set. .Small differences between pathogenic contracted) and non-pathogenic allele. .No correlation with allele- specific % of *DUX4* expression.
[70]	MRSE1	FseI site	No	254 controls ? 25 Asymptomatic 186 FSHD1 individuals 74 FSHD2-(just SMCHD1 mut)	ASCC	Yes; Asymptomatyc carriers not clearly defined	FseI site	Yes for 1–6 repeats carriers No for 7–10 repeats carriers	No for FSHD1; Yes for FSHD2	.Unclear interpretation of data due to a global estimation of methylation as a function of D4Z4 repeat lengths. .Non-penetrant carriers of a DRA included in analysis.
[71]	meDIP BSS	D4Z4-5P; 22 CpGs D4Z4-MID; 31 CpGs D4Z4-3P; 14 CpGs	No	20 Controls 29 Asymptomatic 37 FSHD1 9 FSHD2 8 FSHD1 4FSHD2 1Asymptomatic 7 Controls	Not reported	Yes; Asymptomatyc carriers with methylation levels not different than controls	D4Z4-5’ only No MID or 3’	No	No	.Limited BSS analysis. No correlation with the global number of residual repeats.
[72]	BSS MRSE1	DUX4-PAS FseI site	Yes No	51 Controls 2Asymptomatic 44FSHD1 17 FSHD2 (just SMCHD1 mut)	ASCC	Yes; Asymptomatyc carriers not clearly defined No	DUX4-PAS: pLAM only No FseI	Yes	Yes	.Arbitrary selection of just one (number 6) CpG for statistical analysis. .Non-penetrant carriers of a DRA included in analysis.
[73]	MRSE1+ MRSE2 BSS+ MRSE1+ MRSE2	FseI site BsaAI site 4qA; 56 CpGs or 4qAL; 30 CpGs	No Yes Yes	122P+110: 88 FSHD1 P + 47FSHD1R 34 FSHD2P+ 2 FSHD 2 R 61 Asymptomatic R 1 Control 11FSHD1 8 Asymptomatic	CCEF	Assumed: beyond the purpose of the paper. Asymptomatyc carriers with methylation levels not different than index cases	FseI site 4qA: pLAM	No	No	-Limited BSS analysis
[74]	BSS +NGS	D4Z4-5P; 22 CpGs D4Z4-MID; 31 CpGs D4Z4-3P; 14CpGs DR1; 31CpGs 4aAint;14 CpGs 4qAext;42 CpGs	No Yes	10 Controls 29 FSHD1 10 FSHD2-(just SMCHD1 mut)	Not reported	Yes Asymptomatyc carriers not included in analysis	D4Z4-5’ only	No	No	.Non-penetrant carriers of a DRA not included in analysis.

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
