# Peer review of "Does DNA Methylation Matter in FSHD?"

_genes, 2020, doi:10.3390/genes11030258_

Round 1
Reviewer 1 Report
The authors review different studies of DNA methylation performed on the 4q35 D4Z4 repeat array in patients affected with FSHD, and highlight the difficulties at correlating these data with disease status and evolution. In addition, the authors considered cases of reduced D4Z4 methylation not associated with muscle pathologies such as Bosma Arhinia or ICF syndrome. They finally review data on D4Z4 epigenetic reprogramming in human embryonic and iPS cells. Their general conclusion is that D4Z4 DNA methylation is reflecting chromatin structure but no indicator of its function in FSHD muscle cells.
This review is well written and highlights the complexity of understanding patient heterogeneity in FSHD.
Comments
Abstract
The abstract is misleading. A decrease in DNA methylation at the D4Z4 repeats in 4q35 is not sufficient to cause FSHD. This epigenetic feature has to come in addition to the permissive genetic condition that is the presence of a DUX4 gene on a 4qA allele providing a polyadenylation signal (PAS) : this signal allows for stabilization of the transcribed DUX4 mRNA and its translation to cytotoxic DUX4 protein. This should be mentioned in the abstract, as well as the key concept that chromatin relaxation on the D4Z4 repeat array is thought to lead to cytotoxic DUX4 expression both in FSHD1 (4qA monogenic transmission) and FSHD2 (digenic transmission: 4qA + mutation in chromatin remodeler).
Introduction
Line 36: the text should be corrected since at least one D4Z4 unit (i.e. one DUX4 gene) is required to develop FSHD (Prof. Tupler et al 1996).
The Authors should distinguish FSHD 1 (monogenic: DUX4 gene on 4qA) from FSHD2 (digenic: DUX4 gene on 4qA + loss of function mutation of a gene involved in chromatin structure)
Line 45: define RU here or above
Line 52: the Authors should present the concept of a continuum between FSHD1 and 2
Line 58: Fig 1 mentions DUX4 transcript in a column on the schematic but this is explained neither in the text nor in the legend. Please clarify. Several typos to be corrected on the figure itself: myopathies, healthy.
Lines 70-78: additional chromatin factors (proteins and RNAs) associated with D4Z4 have been described by other researchers than Prof. Tupler’s collaborators and should also be mentioned here: Miller, Jones, Tapscott, Chadwick...
Lines 97, 98: The following statement lacks precision: “presence of a specific telomeric polymorphism, 4qA, whichallows the expression of the most distal copy of the DUX4 gene “
In fact DUX4 gene transcription can occur on any 4q allele, but the PAS present on 4qA allows for DUX4 mRNA polyadenylation, stabilization and translation into a cytotoxic protein.
Appropriate references about DUX4 gene characterization should be provided ( Kowaljow et al 2007; Dixit et al 2007; Lemmers et al 2010)
Line 147: the data obtained by Jones et al on DUX4 mRNA or DUX4 protein expression should be summarized besides the DNA methylation data
The debated role of DNA methylation in FSHD: clinical and families studies.
Line 189: hypomethylated CpGs are not located in the proximal promoter, but where do they map as compared to the D4Z4 enhancers and MARs identified by other groups (Vassetzky, Jones)?
Table 1: The recent publication by Roche et al 2020 of Prof. Magdinier’s group should be included in the table and its discussion
The significance of D4Z4 hypomethylation in FSHD and its implication for clinical counseling.
It is now well established that DNA methylation at CpGs is only one among numerous indicators of chromatin structure on D4Z4 and publications from several research groups have studied alterations of chromatin structure reflected by post-translational histone modifications, or binding of various non-histone proteins and RNAs on the repeat array. Moreover, the DUX4 protein itself induces extremely potent and long-term impact on chromatin opening by genome wide histone acetylation (via p300/CPB; Bosnakovski et al 2019) a phenomenon probably involved in DUX4 normal function i.e. zygote genome activation. Moreover DUX4 transcriptionally activates expression of H3.X and H3.Y histone variants which get incorporated into nucleosomes leading to relaxed chromatin regions on promoters of DUX4 target genes (Resnick et al 2019). Through both nucleosome composition and histone acetylation DUX4 could also contribute to chromatin opening on D4Z4.
The contribution of DUX4 to D4Z4 chromatin structure should be discussed.
Minor comment
There are major issues with words spacing in the text.
Reviewer 2 Report
The review provides a critical and unbiased analysis of the current state of DNA methylation studies in FSHD. Unfortunately, the conclusions are quite trivial and somewhat expected, although this does not undermine the quality of the work. My major concern is that the authors consider DNA methylation in FSHD per se, although the effect of DNA methylation is exerted through direct or indirect changes in chromatin or via altered interaction of transcription factors with methylated DNA. A chapter on this subject will significantly strengthen the review.
Minor remarks:
Many words are fused throughout the text, eg "repeatis" on line 12 etc. etc. The quality of the Figure 1 can be improvedAuthor Response
Please see the attachment.
